# A Ten-Year Trend Analysis of *Fusarium* Mycotoxins in Croatian Maize (2014–2024)

**DOI:** 10.3390/foods14132209

**Published:** 2025-06-23

**Authors:** Nina Kudumija, Tina Lešić, Manuela Zadravec, Ana Vulić, Jelka Pleadin

**Affiliations:** 1Laboratory for Analytical Chemistry, Croatian Veterinary Institute, Savska Cesta 143, 10000 Zagreb, Croatia; kudumija@veinst.hr (N.K.); lesic@veinst.hr (T.L.); pleadin@veinst.hr (J.P.); 2Laboratory for Feed Microbiology, Croatian Veterinary Institute, Savska Cesta 143, 10000 Zagreb, Croatia; zadravec@veinst.hr

**Keywords:** cereals, ELISA, zearalenon, deoxinyvalenol, fumonisin, T-2/HT-2, trend

## Abstract

Maize is one of the most important agricultural crops that has been cultivated in the Republic of Croatia for centuries. *Fusarium* mycotoxins as secondary metabolites of molds that naturally contaminate maize crops can have negative effects on human and animal health, but also on economic aspects. The aim of this study was to monitor the trend of natural occurrence of *Fusarium* mycotoxins in maize crops from Croatia during a period of 10 years, from 2014 to 2024. A total of 1285 maize samples were analyzed for the contents of zearalenone (ZEN), deoxynivalenol (DON), fumonisins B (FUM) and T-2/HT-2 toxin (T-2/HT-2) using validated ELISA methods. A high occurrence of the analyzed mycotoxins was found in all years examined, e.g., for ZEN 15–64%, DON 47–95%, FUM 60–96% and T-2/HT-2 24–89%. In addition, their concentrations ranged from 3.2–10,990 μg/kg, 18–25,000 μg/kg, 29–18,180 μg/kg and 9–595 μg/kg, respectively. The mycotoxin concentrations were found to differ from year to year, with an increasing trend observed for FUM and T-2/HT-2 in terms of concentrations and an increasing trend in the occurrence of all mycotoxins, with the exception of T-2/HT-2.

## 1. Introduction

Maize is the most widely grown cereal in Croatia, and its production is directly influenced by climate and market trends [1]. According to statistical data (period from 2014 to 2024), Croatian maize production averaged 2.82 million tons per year, with a cultivated area of around 300,000 hectares [2]. Iljikić et al. (2019) [3] state that although Croatia is located in the optimal growing area for maize production at the 15–45° northern latitude with relatively fertile soils, differences in production can vary by more than 50% between years, especially due to climatic factors (insufficient and uneven rainfall during the growing season and high temperatures in the phenological phases of flowering and fertilization). In addition, there is a direct link between climatic factors and the contamination of crops with toxic molds, which can subsequently form mycotoxins and represent the most important hazard category in the context of food safety [4,5,6].

The problem of mycotoxins is particularly pronounced in rainy years, when the percentage of mold contamination and the resulting formation of mycotoxins increase significantly [7]. Many types of molds contaminate cereal crops at all stages of production, from growth in the field to harvest, transport and storage [7,8]. Field molds of the genera *Fusarium*, *Alternaria*, *Cladosporium* and *Rhizopus* occur during the growing season and require larger quantities of water, while storage molds of the genera *Aspergillus* and *Penicillium* spp. do not require larger quantities of water, and their formation is strongly influenced by temperature and grain damage [9,10]. Cereals and oilseeds with higher sugar and fat contents are more likely to be contaminated with molds and mycotoxins than cereals with a higher protein content. The Zn content is also a suitable substrate for the formation of mycotoxins [11]. Due to its nutritional composition, maize is a very favorable grain that is constantly exposed to the risk of mold growth. For the biosynthesis of mycotoxins to take place, factors such as moisture, pH, temperature, oxygen content and physical damage to the grain must be taken into account [12].

*Fusarium* molds are among the most important pathogenic molds that can produce mycotoxins in maize crops, which primarily depends on the genetic predisposition of the maize, i.e., the maize variety, the production and cultivation methods and the climatic conditions [13]. Zearalenone, trichothecenes and fumonosins are three classes of mycotoxins that are specific to *Fusarium* molds, and they not only have an impact on the economy but are also associated with diseases in humans and animals, from acute to chronic diseases, and are also thought to have carcinogenic, mutagenic, teratogenic, immunosuppressive and estrogenic effects [14,15].

ZEN belongs to the group of hydroxylated resorcylic acid lactones [11] and is produced by various strains of the genus *Fusarium*, including *F. roseum*, *F. culmorum*, *F. tricinctum* and *F. moniliforme*. Over 150 different metabolites of ZEN are known, of which α-zeranol is the most toxic. ZEN has estrogenic effects, inhibits the formation of follicle-stimulating hormone, increases progesterone concentration and decreases testosterone concentration [11,16]. The biological effect of ZEN manifests itself in premature puberty in children [16], while in animals, the ingestion of ZEN-contaminated feed leads to the development of female characteristics and testicular atrophy in males and premature sexual maturity in females. It also leads to reduced food intake and food refusal, with these effects being most pronounced in pigs [14]. Apart from the above-mentioned oestrogenic effects, it also has hepatotoxic, immunotoxic and genotoxic effects [17].

Furthermore, molds of the genus *Fusarium* can produce trichothecenes, a large group of related mycotoxins that frequently occur as contaminants in the food chain. Based on their functional groups, they are divided into four types from A to D; however, A and B are specific for *Fusarium* molds, while C and D are not synthesized by *Fusarium* molds [18,19]. In addition, the trichothecenes of groups A and B are of greater importance due to their toxicity and natural occurrence [18]. They are characterized by their stability against degradation by environmental influences such as light, temperature and the milling process of the grain. T-2 and HT-2 belong to the trichothecenes of type A. They are produced by *F. langsethiae*, *F. poae*, *F. equiseti*, *F. sporotrichioides* and *F. acumninatum* in a wide temperature range of 0–32 °C with a maximum of productivity below 15 °C and frequently contaminate barley, wheat, maize, oats and rice crops.

T-2 toxin has a strong cytotoxic and immunosuppressive effect and inhibits protein synthesis, leading to secondary disruption of nucleic acids (DNA and RNA), impairment of blood and reproductive function and skin irritation, but also anorexia, vomiting and nausea, weight loss and abortions [14,16,20,21,22]. The HT-2 toxin is a less toxic metabolite of the T-2 toxin. DON belongs to group B of the trichothecene mycotoxins and is most frequently produced by *F. graminearum* and *F. culmorum* at optimum temperatures of 25 to 28 °C and contaminates cereals such as maize, wheat and barley more frequently than oats and rye. It is one of the most important food contaminants and also occurs in the form of derivatives such as the acetylated forms 3-ADON and 15-ADON as well as nivalenol. Ingestion of small amounts of DON leads to health problems in the form of lesions in the digestive tract and changes in the immune system. In animals, acute exposure to DON leads to reduced food intake and vomiting, and chronic exposure leads to reduced growth and oscillations in growth hormone production. In humans, DON causes acute nausea, vomiting, diarrhea and abdominal pain. Compared to other trichothecenes, DON is considered one of the less toxic but has a pronounced mutagenic effect in combination with other mycotoxins [11,14,16].

To date, 28 fumonisins are known and described, which are divided into the four categories A, B, C and P. Besides the group mentioned above, the most common fumonisins in food are FB1, FB2 and FB3, which usually occur together in a ratio of 68:20:12 [23], with fumonisin B1 being the most widespread natural maize contaminant synthesized by *F. verticillioides* and *F. proliferatum* [14]. The literature shows that fumonisins can be associated with esophageal cancer and liver and kidney disease in humans. In animals, particularly horses and donkeys, they can cause leukoencephaly and other physical damage such as pulmonary edema and hydrothorax in pigs [14,23,24]. In addition, the presence of fumonisins in feed can have a negative impact on meat quality, especially in the context of increased fat content, which in turn can lead to major economic losses for producers [25].

Since maize is a cereal that plays an important role in the global food system through direct consumption (13%) and in animal feed (61%) [13,26] and is a suitable matrix for the development of *Fusarium* molds and the subsequent formation of mycotoxins due to its chemical composition, genetic predisposition and increasingly frequent climatic changes [13,27,28], the aim of this study was to monitor the trend of occurrence of regulated *Fusarium* mycotoxins on maize crops in the Republic of Croatia during a 10-year (2014–2024) period. To the best of our knowledge, this study represents the first assessment of the trend in the occurrence of *Fusarium* mycotoxins in Croatia over the course of a decade.

## 2. Materials and Methods

### 2.1. Sampling and Sample Preparation

During the period of monitoring *Fusarium* mycotoxins in maize samples in the years 2014–2024, a total of 1245 maize samples were analyzed for ZEN, 919 for DON, 703 for FUM and 637 for T-2/HT-2 toxin presence. The maize samples were collected from different fields on the Croatian mainland at 45°10′′ (N) width, 15°30′′ (E) length, i.e., northern, central and eastern Croatia specific for maize cultivation. Sampling was carried out in accordance with Commission Regulation (EC) No. 401/2006 [29]. All maize samples were thoroughly ground in the analytical mill (Cylotec 1093, Tecator, Höganäs, Sweden) to achieve a particle size of 1.00 mm and stored at 4 °C prior to analysis. All samples were analyzed for mycotoxins within 72 h.

### 2.2. Analysis of Mycotoxins

A Ridascreen ZEN, DON, FUM and T-2/HT-2 toxin kits for competitive enzyme-linked immunosorbent assays (ELISA) were provided by R-Biopharm (Darmstadt, Germany). Each kit contained a 96-well microtiter plate coated with specific antibodies; standard solutions of ZEN (at concentrations of 0, 0.05, 0.15, 0.45, 1.35 and 4.05 μg/kg), DON (at concentrations of 0, 3.7, 11.1, 33.3 and 100 μg/kg), FUM (at concentrations of 0, 25, 74, 222, 666 and 2000 μg/kg), T-2/HT-2 toxin (at concentrations of 0, 1, 3, 6, 12 and 36 μg/kg); peroxidase-conjugated ZEN/DON/FUM/T-2/HT-2; anti-ZEN/DON/FUM/T-2/HT-2 antibody; substrate/chromogen solution (urea peroxide/tetramethylbenzidine); stop solution and wash/dilution buffer. After grinding, 5 g of each sample were extracted with 25 mL distilled water for DON analysis and with 25 mL methanol/water (70/30) for ZEN, FUM and T-2/HT-2 toxin analysis. Extraction was performed by vigorous overhead shaking (multi RS-60, Biosan, Riga, Latvia) for 10 min and centrifugation (Universal 320R Hettich, Tuttlingen, Germany) for 10 min at 3500 rpm at room temperature. The supernatants obtained were diluted accordingly and used for ELISA tests. The ELISA method was performed with ChemWell (Awareness Technology 2910, Inc., Palm City, FL, USA) until 2022 and thereafter with ThunderBolt (GSD, Davis, CA, USA). For both devices, the absorbance of the measurement was 450 nm. The ELISA tests were performed according to the manufacturer’s instructions. The mycotoxin concentrations in the samples were calculated using the software Rida Soft win (version 1.100.0.0202), R-Biophram (Darmstadt, Germany). Mycotoxin standards and other necessary chemicals were purchased from Sigma Aldrich Chemie GmbH (Steinheim, Germany). The aforementioned ELISA methods used to determine mycotoxin concentrations using the ChemWell device were validated as previously described [21,30,31]. By using the new instrument after 2022, the methods ware revalidated, and the new results for limits of detection (LOD), limits of quantification (LOQ) and recoveries for ZEN, DON, FUM and T-2/HT-2 are shown in Table 1, and structural chemical formulas [32] are shown in Table 2. Quality control was performed on each batch of all samples tested by analyzing various maize reference materials (RM) ordered from FAPAS (SandHutton, York, UK).

### 2.3. Meteorological Data

Data of the weather parameters (temperatures and precipitation) in Croatia during the ten-year period (2014–2024) for the maize growing season, i.e., from the sowing phase to the flowering and harvesting (May–September), were obtained from the Croatian Meteorological and Hydrological Service [33,34].

### 2.4. Statistical Analysis

The statistical analysis was performed using SPSS Statistics Software 22.0 (SPSS Statistics, NYIBM, 2013, Sankt Ingbert, Germany). The differences between the sample groups were determined using the Kruskal–Wallis test with statistical significance set at 95% (*p* = 0.05). The correlation between mycotoxin levels was assessed using Spearman’s rank correlation test, due to the absence of both univariate and bivariate normality.

## 3. Results and Discussion

Many studies pointed to the ubiquitous presence of mycotoxins in different foodstuffs as one of the major health and economic problems [8,13,14,19,27]. As there is no single process that can remove most mycotoxins from cereals without compromising price and nutritional value, it remains one of the major challenges. Moreover, we are witnessing a climate change that is significantly affecting the whole world, with negative impacts on health, ecosystems, infrastructure and food production [35].

The concentrations of mycotoxins observed in this study during the period 2014–2024 were in the ranges of 3.2–10,990 μg/kg for ZEN, 18–25,000 μg/kg for DON, 29–18,180 μg/kg for FUM and 9–595 μg/kg for T-2/HT-2 toxin, respectively (Table 3). Figure 1 shows the mean concentrations (μg/kg) of *Fusarium* mxcotoxins over the 2014–2024 year period with trend lines for each mycotoxin. In 2014 and 2015, the average highest concentrations for ZEN were 559 ± 3302 and 4621 ± 5168 μg/kg and for DON 2478 ± 3302 and 4621 ± 5186 μg/kg, respectively. In addition, the two mentioned years differ statistically significantly (*p* < 0.05) from the other observed years in terms of ZEN and DON. Also in these two years, the highest percentages of 30% and 45% for ZEN and 34% and 42% respectively for DON samples exceeding the maximum levels (ML) of 350 μg/kg for ZEN and 1500 μg/kg for DON were observed [36,37,38].

Hajnal et al. (2023) [8] also mention 2014 as the year with the highest recorded precipitation values in Serbia, and certain mean values ranged from 15 to 2596 μg/kg for ZEN, while for DON these values ranged from 428 to 16,350 μg/kg. The same authors give mean values for the Republic of Croatia of 810 ± 858 μg/kg for ZEN in 2014 and 1519 ± 1754 μg/kg and 1998 ± +2517 μg/kg and 3711 ± 2710 μg/kg for DON in 2015. They also associate the results obtained with high precipitation and high temperatures in the months characteristic for maize flowering and ripening. The observed occurrence of the analyzed mycotoxins varied over the years and was 15–64% for ZEN, 47–95% for DON, 60–96% for FUM and 24–89% for T-2/HT-2.

Figure 2 shows the trend of Fusarium mycotoxins occurrence in Croatia in the period 2014–2024, and a different occurrence of the analyzed mycotoxins was observed, with an increasing trend for ZEN, DON and FUM compared to a decreasing occurrence of T-2/HT-2 toxins. In a study conducted by Czembor et al. in 2015 [39] on maize samples collected in Poland in 2011 and 2012, the occurrence of ZEN was found in 43% and DON in 67% of the samples. The occurrence of ZEN is associated with the occurrence of Gibberella ear rot (GER) in maize crops caused by *F. Graminearum*, but also *F. Culmorum*, which occurs in areas with moderate temperatures and increased rainfall during the growing season. The global presence of mycotoxins is also indicated by the studies of Schatzmayer and Streit (2013) [40], in which a total of 19,757 feed and grain samples were analyzed, and the prevalence of ZEN was determined to be 22% in Northern Europe, 13% in Eastern Europe, 25% in Central Europe and 21% in Southern Europe, while the prevalence of DON was 64%, 33%, 58% and 51% respectively. The same study also states that the prevalence of ZEN is 56% in North Asia, 25% in South Asia, 28% in Africa and 20% in Oceania, while the prevalence of DON is 78%, 21%, 66%, 34%, 68% and 16% respectively. The above-mentioned occurrence of mycotoxins varies greatly depending on the geographical area, while in this study, variability in the occurrence of mycotoxins was observed in one geographical area depending on the observed year. The authors also explain that weather conditions are the most important factor for mold growth and the subsequent formation of mycotoxins in maize crops. This is proven by the analysis of maize in America in 2010 with a high occurrence of ZEN of 43% and of DON of 92% with an average concentration of 165 μg/kg for ZEN and 1366 μg/kg for DON. In addition, the months of the maize harvest (September-October 2010) were extremely cold and rainy, but September and October 2011 were warmer than average with below-average precipitation, so that the incidence of ZEN was 16% and of DON was 62%. Pleadin et al. [12] point out that 2010 in Croatia was an extremely rainy year with lower average temperatures in the months of the maize harvest, which led to occurrences of ZEN and DON of 85% and 87.5%, respectively.

As far as *Fusarium* mycotoxins are concerned, maize is the crop most frequently contaminated with FUM. In this study, the lowest mean FUM concentrations of 820 ± 1280 μg/kg were determined in 2017 and the highest of 3340 ± 4550 μg/kg in 2022. In the same year, the highest percentage (25%) of samples exceeding the ML value of 4000 mg/kg for FUM was found, while this percentage was significantly lower in the other years observed (0–9%). The occurrence of FUM was also extremely high over the years, especially in 2020, when it amounted to 96%. In addition, increased FUM concentrations were observed in 2022, 2023 and 2024 when temperatures were above 23 °C during the months of maize flowering, while literature data indicate that the optimal temperature for FUM biosynthesis is 15 to 25 °C [41]. Czembor et al. (2015) [39] reported a 100% incidence of FUM in the range of 59.68–1190.33 μg/kg. According to Schatzmayr and Streit 2013 [40], the global incidence of FUM is 51% in Central Europe, 70% in Southern Europe, 55% in Asia, 10% in Oceania, 72% in Africa, 48% in North America and 77% in South America. Hajnal et al. 2023 [8] report 89% of positive maize samples for the period from 2018 to 2021, and certain concentrations were in the range of 24–13,800 μg/kg. Carbas et al. in 2025 [42] in Portugal determined from 150 maize samples (sampled from 2018 to 2020) a mean FUM concentration of 1275 μg/kg (ranging from 62.5 to 5915 μg/kg), which corresponds to similar values determined in this study for the period from 2014 to 2021 and 2024. In addition, Oliviera et al. in 2017 [43] reported 2204 mg/kg as the mean FUM concentrations in maize from Brazil. This is similar to our results from 2023, in which a mean FUM value of 1830 μg/kg was determined. Fluctuations in FUM concentrations correlate with climatic conditions, agricultural practices and maize variety [42].

Among all the mycotoxins analyzed in this study, the lowest concentrations were found for T-2/HT-2, ranging from 9 to 595 μg/kg. The highest incidence of the mentioned mycotoxin was 89% for 2014, and the lowest mean concentration of T-2/HT-2 of 14.2 μg/kg, with a range of 9 to 52 μg/kg, was also found in 2014. It is interesting to note that the lowest occurrence of T-2/HT-2 was found in 2017 at 24% but also the highest average concentration of 77.3 μg/kg. This year was characterized by a rainfall of over 200 mm and temperatures below 17 °C during the maize harvest. Over the years, it was found that in 2024, 9% of the samples exceeded the ML values for T-2/HT-2 toxin, while in the other years, they were between 0 and 7%. The occurrence of T-2/HT-2 in 90 samples of maize from Serbia from 2012 was 53% with a mean maximum concentration of 209 μg/kg, indicating that the drought in 2012 favored the growth of some *Fusarium* species, especially for the producer of FUM, as the occurrence of FUM in these samples was 100% [44]. In previous studies in Croatia, the occurrence of T-2/HT-2 in maize samples in 2017 and 2018 was 26.8%, with concentrations between 15.6 and 332.3 μg/kg [45], and in the period from 2018 to 2021, it was 50%, with a concentration range of 11–407 μg/kg [8]. In this study, it was found that of all the maize samples analyzed, 11% exceeded the maximum permitted level for ZEN, 12% for DON, 7% for FUM and 3% for T-2/HT-2. The 2020 study by Topi et al. [46] in Albania found that 31% of the maize samples exceeded the maximum level for FUM. Kirinčić et al. In 2015 [47] found that 40% of the cereal samples analyzed in Slovenia contained mycotoxins, and 2.4% of the samples exceeded the EU maximum level; the most frequently co-occurring mycotoxin in maize was FUM-DON-ZEN. Tima et al. In 2015 [48] also found that *Fusarium* mycotoxins are widespread plant contaminants in Middle and Eastern Europe and with ELISA methods, detected DON in 86%, ZEN in 41% and T-2 in 55% of the maize samples analyzed from Hungary.

It should also be noted that according to data from the Croatian Meteorological and Hydrological Service [33,34], all observed years, not only the monitored period of May–September, were considered extremely warm, and 2014 was particularly characterized by a large amount of precipitation. Figure 3a shows the temperatures, and Figure 3b shows the precipitation during a ten-year period in the Republic of Croatia, in the months that are characteristic for the planting, flowering and harvesting of maize. In May, during the maize cultivation phase, the average air temperatures were between 13.6 °C (2019) and 19.2 °C (2020) with an average precipitation of 60 mm (2022) to 170 mm (2019). In July, during the maize flowering period, average temperatures ranged between 21.5 °C (2014) and 25 °C (2024), and precipitation ranged from 33 mm (2023) to about 116 mm (2014). In the maize harvest phase (during September), average air temperatures ranged from 15.6 °C (2017) to 19.9 °C (2023), with average precipitation around 200 mm in 2014, 2017 and 2022 [27,28].

When monitoring the occurrence of *Fusarium* mycotoxins in the period 2014–2024 in the Republic of Croatia, statistically significant differences were found in relation to individual mycotoxins depending on the year of harvesting, which can be directly linked to weather conditions recorded that year. The literature shows that the effects of climate change are likely to increase the occurrence of mycotoxins in general and that temperature, precipitation and wind have the greatest impact during the flowering phase. However, these weather influences vary by region, and it is assumed that higher precipitation and higher temperatures in some regions lead to favorable climatic conditions for the development of *Fusarium* mold species [5]. Qu et al. in 2024 [6] stated that the occurrence and development of *Fusarium* mycotoxin contamination will change with climate change, especially due to variations in temperature, precipitation and carbon dioxide concentration, and declared that the principles of good agricultural practice must be applied to produce safe and healthy food. Mielniczuk and Skwarylo-Bednarz in 2020 [49] noted that good agricultural practice is the best defense for controlling contamination of cereals and maize with *Fusarium* mycotoxins, but that fluctuations in weather conditions can significantly reduce the effectiveness of crop protection methods against *Fusarium spp.* infections and accumulation of mycotoxins in cereals.

The results of this study show that determined mycotoxin concentrations differed from year to year, with an increasing trend observed for FUM and T-2/HT-2 in terms of concentrations and also an increasing trend in the occurrence of all mycotoxins, with the exception of T-2/HT-2 toxins. Further, a weak negative cooccurrence (at the 0.01 level) was found between ZEN and T2/HT2 (rs = −0.292), while very weak negative cooccurrences were found between DON and FUM as well as DON and T2/HT2 with rs = −0.116 and rs = −0.126, respectively. In contrast, a weak positive cooccurrence was found between DON and ZEN (rs = 0.178). In addition, the data on the worldwide occurrence of the mycotoxin mentioned above vary depending on the climate zone of the individual regions [22].

## 4. Conclusions

Maize is one of the most important cereals in the world for human and animal nutrition, but it is also suitable for the development of molds and the subsequent formation of mycotoxins. In this study, covering the period from 2014 to 2024, a high incidence of *Fusarium* mycotoxins (ZEN, DON, FUM, T-2/HT-2) was found in maize crops. In addition, an increasing trend in the concentrations of FUM and T-2/HT-2 and an increasing trend in the occurrence of ZEN, DON and FUM was observed. Weather conditions, especially daily temperatures and precipitation during the biological growth stages of maize, play an important role in mycotoxin formation. In view of the problems associated with climate change, it is necessary to cultivate maize varieties that are resistant to molds in order to reduce their negative impact on health and the economy.

## Figures and Tables

**Figure 1 foods-14-02209-f001:**
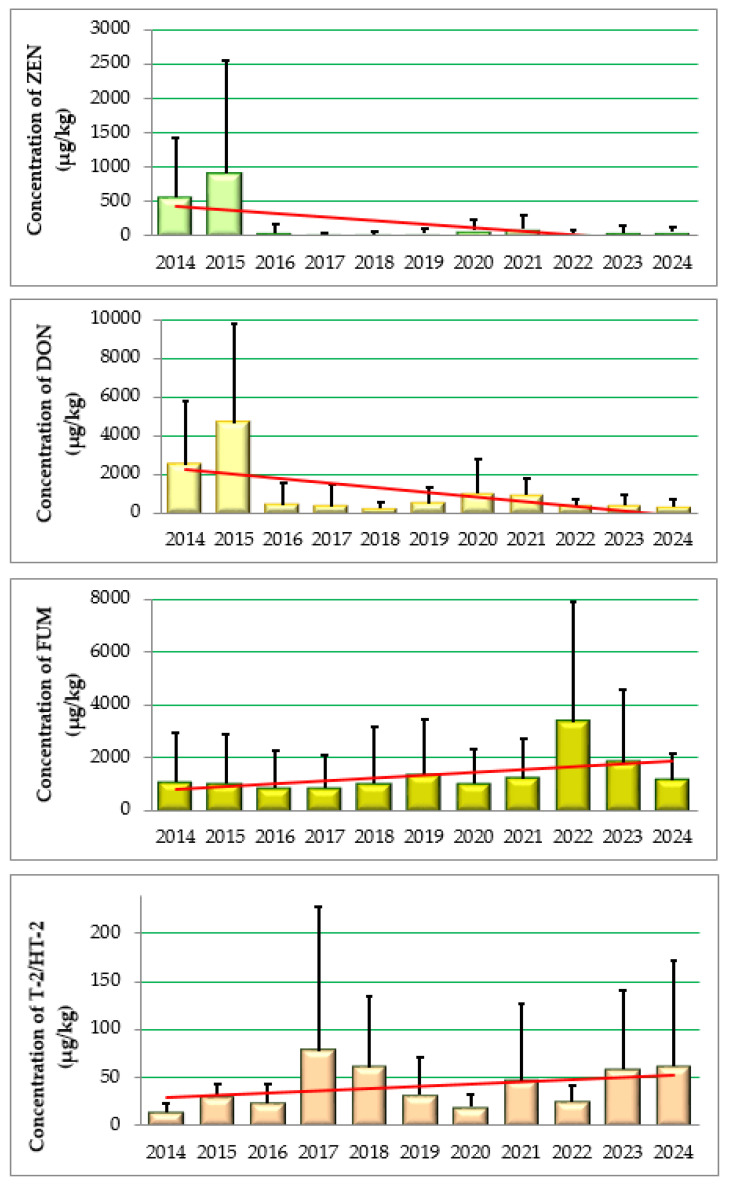
Mean concentrations (μg/kg) of *Fusarium* mxcotoxins over the 2014–2024 years with equations of trend lines (ZEN: y = −52.36x + 484.47; DON: y = −238.4x + 2469.8; FUM: y = 108.6x + 672.22; T-2/HT-2: y = 2.24x + 27.18).

**Figure 2 foods-14-02209-f002:**
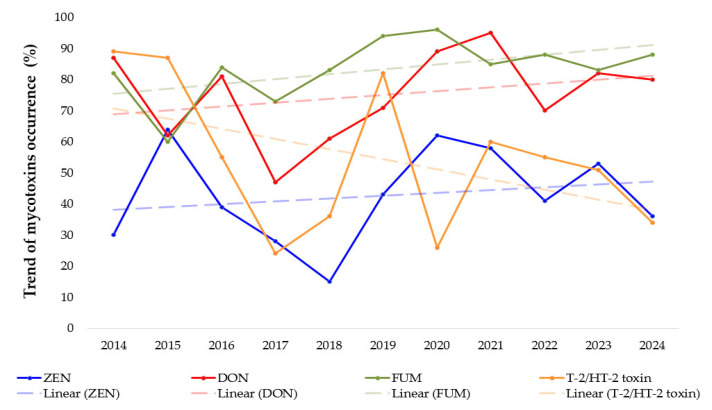
Trend of *Fusarium* mycotoxins occurrence (%) in period of 2014–2024 with linear regression with equations of lines (ZEN: y = 0.9x + 37.24; DON: y= 1.24x + 67.58; FUM: y = 1.55x + 73.95; T-2/HT-2: y = −3.25x + 73.93).

**Figure 3 foods-14-02209-f003:**
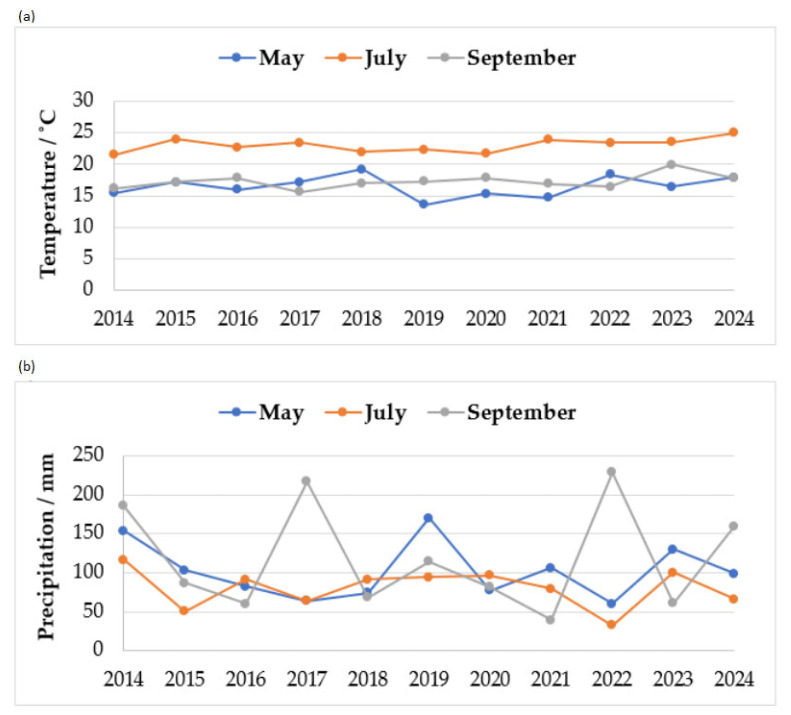
Weather conditions during a ten-year period in the Republic of Croatia (2014–2024): (**a**) temperatures (°C) and (**b**) the precipitation (mm) [27,28].

**Table 1 foods-14-02209-t001:** Limits of detection, limits of quantification and recoveries for ZEN, DON, FUM and T-2/HT-2 after applying ThunderBolt device and revalidation of methods.

	ZEN	DON	FUM	T-2/HT-2
LOD ^a^ (μg/kg)	1.9	17.7	29.0	8.9
LOQ ^a^ (μg/kg)	3.1	24.2	40.0	14.8
Recovery ^a^ (%)	84.0	102.0	113.0	73.0

LOD—limit of detection; LOQ—limit of quantification; ZEN—zearalenone; DON—deoxynivalenol; FUM—fumonisin B; T-2/HT-2—T-2/HT-2 toxin; ^a^ these validation parameters are calculated from 6 replicates.

**Table 2 foods-14-02209-t002:** Chemical strucures of Fusarium mycotoxins [32].

Mycotoxin Name	Chemical Formula
Zearalenon (ZEN)	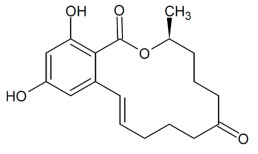
Deoxynivalenol (DON)	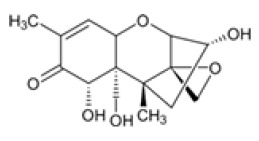
Fumonisin (FUM)	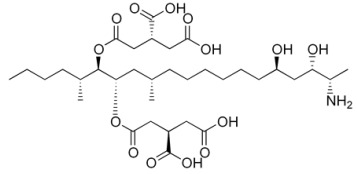
T-2 toxin (T-2)	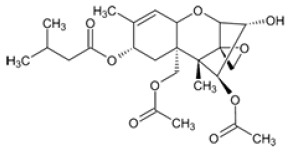
HT-2 toxin (HT-2)	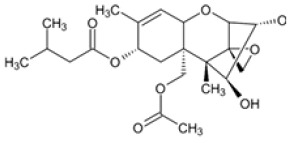

**Table 3 foods-14-02209-t003:** *Fusarium* mycotoxins in Croatia in the period 2014–2024.

	2014	2015	2016	2017	2018	2019	2020	2021	2022	2023	2024
ZEN											
N	158	144	97	109	103	121	95	96	121	105	96
N of positives	110	92	38	30	15	52	59	56	49	56	35
N > ML	47	65	2	0	1	3	6	6	0	1	1
% > ML	30	45	2	0	1	2	6	6	0	1	1
Min-Max	3.2–5552	3.2–10,990	3.2–1084	3.2–113	3.2–479	3.2–658	3.2–1240	3.2–1170	3.2–193	3.2–728	3.2–361
Median	23 ^ac^	204 ^c^	9 ^b^	9 ^b^	9 ^b^	9 ^b^	9 ^b^	12 ^d^	16 ^d^	11 ^d^	28 ^a^
DON											
N	90	105	73	93	82	82	70	75	93	82	74
N of positives	78	65	59	44	50	58	62	71	65	67	59
N > ML	31	44	1	2	0	5	9	10	2	3	2
% > ML	34	42	1	2	0	6	13	13	2	4	3
Min-Max	52–18,330	66–25,000	25–8620	39–1782	18–1232	26–4687	78–9923	43–5133	36–1959	21–2917	20–2101
Median	899 ^a^	3120 ^c^	221 ^b^	220 ^b^	101 ^d^	166 ^b^	523 ^e^	610 ^e^	201 ^b^	198 ^b^	139 ^b^
FUM											
N	71	81	51	80	63	68	50	48	65	69	57
N of positives	58	49	43	58	52	64	48	41	57	57	50
N > ML	6	4	3	3	3	4	3	2	16	6	0
% > ML	8	5	6	4	5	6	6	4	25	9	0
Min-Max	42–8900	50–11,100	29–6800	32–5202	29–13,800	29–11,530	29–5920	60–6330	39–1942	41–18,180	71–3820
Median	189 ^a^	232 ^a^	171 ^a^	167 ^a^	343 ^a^	654 ^b^	593 ^ab^	416 ^ab^	1.166 ^c^	818 ^b^	750 ^b^
T-2/HT-2											
N	54	43	56	76	90	101	50	47	40	45	35
N of positives	48	39	31	18	32	83	13	28	22	23	12
N > ML	0	0	0	3	4	4	0	3	0	3	3
% > ML	0	0	0	4	4	4	0	6	0	7	9
Min-Max	9–52	9–75	9–97	10–595	11–323	10–282	11–50	11–407	5–66	10–347	11–410
Median	14 ^a^	32 ^ac^	17 ^b^	15 ^b^	36 ^d^	18 ^b^	13 ^b^	22 ^bcd^	20 ^bc^	20 ^bcd^	18 ^bcd^

ZEN—zearalenone; DON—deoxynivalenol; FUM—fumonisin B; T-2/HT-2 (T-2/HT-2 toxin); N: number of total samples; N of positives: number of contaminated—positive samples (>limit of detection); N > ML: number of samples exceeding the maximum level of certain mycotoxins; % > ML: percentage of positive samples exceeding the maximum level of certain mycotoxins; Min-Max: minimum and maximum concentrations (μg/kg); median: median concentration of samples > limit of detection (μg/kg); ^a–e^ values within a row with no common superscript difer significantly (*p* < 0.05).

## Data Availability

The original contributions presented in this study are included in the article. Further inquiries can be directed to the corresponding author.

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
