# Peer review of "A Ten-Year Trend Analysis of Fusarium Mycotoxins in Croatian Maize (2014–2024)"

_foods, 2025, doi:10.3390/foods14132209_

Round 1
Reviewer 1 Report
Comments and Suggestions for Authors
Lines 86 to 90 detail the potential harmful effects of fumonisins on animal health, alongside their effects on human health. To maintain consistency, it would be beneficial to include similar descriptions for the other mycotoxins mentioned earlier.
Line 240: Shouldn't it be µg/kg instead mg/kg?
Data in this paper is crucial, and annual monitoring of mycotoxins is essential. Additionally, it would be valuable that the authors discuss the obtained values in relation to existing regulations and recommendations, as this would enhance the depth of the work.
The work needs revisions, which I would categorize as between minor and major.
Author Response
Reviewer #1:
COMMENT 1: Lines 86 to 90 detail the potential harmful effects of fumonisins on animal health, alongside their effects on human health. To maintain consistency, it would be beneficial to include similar descriptions for the other mycotoxins mentioned earlier
RESPONSE: The negative effects of certain mycotoxins on human and animal health have been added (lines: 63-67; 81-83; 90-93)
COMMENT 2: Thank you for your comment. Line 240: Shouldn't it be µg/kg instead mg/kg?
RESPONSE: I apologise for the mistake, instead „mg/kg“ I should write „µg/kg“ (line 269)
COMMENT 3: Data in this paper is crucial, and annual monitoring of mycotoxins is essential. Additionally, it would be valuable that the authors discuss the obtained values in relation to existing regulations and recommendations, as this would enhance the depth of the work
RESPONSE: For the sake of clarity, two new rows per mycotoxin have been added to Table 2, showing the number of samples and the percentage of samples per year that exceed the regulatory maximum level. In addition. In Chapter 3 Results & discussion, provides a textual explanation of the data shown in Table 2 in connection with the existing regulations and recommendations. (rows: 183-186; 242-245; 272-274; 280-282)
The authors are indebted to our esteemed Reviewer for their most helpful suggestions and comments.
Reviewer 2 Report
Comments and Suggestions for Authors
The paper entitled ‘A Ten-Year Trend Analysis of Fusarium Mycotoxins in Croatian Maize (2014-2024)’ is interesting. I have no specific concern but it is mandatory to correct the manuscript in these point:
- Please add a table to show the chemistry information of these Fusarium mycotoxins.
- Weather conditions, especially daily temperatures and precipitation during the biological growth stages of maize, play an important role in mycotoxin formation. These still less a relationship between daily temperaturesand mycotoxin
- The discussion of the increase and decrease rules of different mycotoxins in this paper is not deep enough, and more literature needs to be supplemented.
Author Response
Reviewer #2:
COMMENT 1: Please add a table to show the chemistry information of these Fusarium mycotoxins
RESPONSE: Table 2 has been added, listing the chemical formulae of the Fusarium mycotoxins
COMMENT 2: Weather conditions, especially daily temperatures and precipitation during the biological growth stages of maize, play an important role in mycotoxin formation. These still less a relationship between daily temperaturesand mycotoxin
RESPONSE: Thank you for you comment. New references related to climate change have been added (lines: 309-322)
COMMENT 3: The discussion of the increase and decrease rules of different mycotoxins in this paper is not deep enough, and more literature needs to be supplemented.
RESPONSE: Thank you for your helpful comment. New references are added (lines: 282-289)
The authors are indebted to our esteemed Reviewer for their most helpful suggestions and comments.
Reviewer 3 Report
Comments and Suggestions for Authors
This paper analyzed 1,285 corn samples collected in the Republic of Croatia from 2014 to 2024 for four fungal toxins: ZEN, DON, FUM, and T-2/HT-2. The paper also investigated the relationship between the toxins' trends and weather conditions. While the paper is interesting, the analysis is relatively simple and has the following issues:
1. Line 102, what does "od" mean?
2. Were no parallel experiments conducted in Table 1? The data do not make it clear whether parallel experiments were performed.
3. The data on mycotoxins in lines 161–176 do not align with the data in the abstract. This requires clarification and rigorous verification of all data throughout the paper.
4. The paper only analyzed the detection rates of mycotoxins and did not conduct analyses such as determining whether limit values were exceeded, necessitating further refinement.
5. The results and discussion sections lack a comprehensive comparative analysis. This is the first analysis of Fusarium mycotoxins in the Republic of Croatia; however, other countries have relevant data on these toxins in corn.
6. The article reports that the concentrations of mycotoxins, such as ZEN and DON, reached 10,990 and 25,012 μg/kg, respectively. These concentrations far exceed the linear range of the ELISA kit. How can such high concentrations be accurately measured?
Author Response
Reviewer #3:
COMMENT 1: Line 102, what does "od" mean?
RESPONSE: Thank you for you comment, I apologise for the mistake, instead „od“ I should write „of“. (line 117)
COMMENT 2: Were no parallel experiments conducted in Table 1? The data do not make it clear whether parallel experiments were performed.
RESPONSE: Validation parameters are calculated from 6 replicates (line 168)
COMMENT 3: The data on mycotoxins in lines 161–176 do not align with the data in the abstract. This requires clarification and rigorous verification of all data throughout the paper.
RESPONSE: Thank you for your helpful comment. I apologise for the mistake and data are corrected (lines 178-179)
COMMENT 4: The paper only analyzed the detection rates of mycotoxins and did not conduct analyses such as determining whether limit values were exceeded, necessitating further refinement
RESPONSE: For the sake of clarity, two new rows per mycotoxin have been added to Table 2, showing the number of samples and the percentage of samples per year that exceed the regulatory maximum level. In addition. In Chapter 3 Results & discussion, provides a textual explanation of the data shown in Table 2 in connection with the existing regulations and recommendations. (rows: 183-186; 242-245; 272-274; 280-282)
COMMENT 5: The results and discussion sections lack a comprehensive comparative analysis. This is the first analysis of Fusarium mycotoxins in the Republic of Croatia; however, other countries have relevant data on these toxins in corn.
RESPONSE: Thank you for your helpful comment. New references are added (lines: 282-289)
COMMENT 6: The article reports that the concentrations of mycotoxins, such as ZEN and DON, reached 10,990 and 25,012 μg/kg, respectively. These concentrations far exceed the linear range of the ELISA kit. How can such high concentrations be accurately measured?
RESPONSE: Since we analyzed mycotoxins according to the instructions in the kit, at concentrations that fell outside the curves, we applied additional dilution and included the dilution factor in the final result.
The authors are indebted to our esteemed Reviewer for their most helpful suggestions and comments.
Reviewer 4 Report
Comments and Suggestions for Authors
Line 25-26: Reformulate the first sentence, it is not correct to introduce although at beginning of a sentence.
Line 33-34: Add scientific works that study that influence in mycotoxins occurrence.
Line 37: Specify the type of moulds are influencing the mycotoxins levels at different stages of pre and post-harvest of maize crop.
Line 38-40: introduce scientific works to corroborate the sentence, specifically which nutritional compounds may influence mould growth.
Line 102: correct the mistake
Line 114, 189: Provide italic format the genera ‘Fusarium’
At results, provide the percentage of samples exceeds the legal limits for each mycotoxin found over 10 years in Croation, and point out the main causes of occurrence.
Author Response
Reviewer #4:
COMMENT 1: Line 25-26: Reformulate the first sentence, it is not correct to introduce although at beginning of a sente
RESPONSE: Thank you for the comment, I've corrected the sentence (lines 25-26)
COMMENT 2: Line 33-34: Add scientific works that study that influence in mycotoxins occurrence
RESPONSE: New references related to influence in mycotoxins occurrence have been added (line: 35)
COMMENT 3 Specify the type of moulds are influencing the mycotoxins levels at different stages of pre and post-harvest of maize crop
RESPONSE: Specific type of moulds influencing the mycotoxins at different stages of pre and post harvesting are described in lines 39 - 43
COMMENT 4: introduce scientific works to corroborate the sentence, specifically which nutritional compounds may influence mould growth
RESPONSE: In lines 43-46 are described which nutritional compaunds may influence mould growth.
COMMENT 5: Line 102: correct the mistake
RESPONSE: Thank you for you comment, I apologise for the mistake, instead „od“ I should write „of“. (line 117)
COMMENT 6: Line 189: Provide italic format the genera ‘Fusarium’
RESPONSE: Thank you for the comment, I apologise for the mistake (line 217)
COMMENT 7: At results, provide the percentage of samples exceeds the legal limits for each mycotoxin found over 10 years in Croation, and point out the main causes of occurrence
RESPONSE: For the sake of clarity, two new rows per mycotoxin have been added to Table 2, showing the number of samples and the percentage of samples per year that exceed the regulatory maximum level. In addition. In Chapter 3 Results & discussion, provides a textual explanation of the data shown in Table 2 in connection with the existing regulations and recommendations. (rows: 183-186; 242-245; 272-274; 280-282)
The authors are indebted to our esteemed Reviewer for their most helpful suggestions and comments.